# Risk Prediction for Early Chronic Kidney Disease: Results from an Adult Health Examination Program of 19,270 Individuals

**DOI:** 10.3390/ijerph17144973

**Published:** 2020-07-10

**Authors:** Chin-Chuan Shih, Chi-Jie Lu, Gin-Den Chen, Chi-Chang Chang

**Affiliations:** 1Institute of Medicine, Chung Shan Medical University, Taichung 40201, Taiwan; tracy@usmg.com.tw (C.-C.S.); gdchentw@hotmail.com (G.-D.C.); 2General Administrative Department, United Safety Medical Group, New Taipei City 24205, Taiwan; 3Deputy Chairman, Taiwan Association of Family Medicine, Taipei 24200, Taiwan; 4Graduate Institute of Business Administration, Fu Jen Catholic University, New Taipei 24205, Taiwan; 059099@mail.fju.edu.tw; 5Artificial Intelligence Development Center, Fu Jen Catholic University, New Taipei 24205, Taiwan; 6Department of Obstetrics and Gynecology, Chung Shan Medical University Hospital, Taichung 40201, Taiwan; 7School of Medical Informatics, Chung Shan Medical University & IT office, Chung Shan Medical University Hospital, Taichung 40201, Taiwan

**Keywords:** early chronic kidney disease, machine learning, risk prediction

## Abstract

Developing effective risk prediction models is a cost-effective approach to predicting complications of chronic kidney disease (CKD) and mortality rates; however, there is inadequate evidence to support screening for CKD. In this study, four data mining algorithms, including a classification and regression tree, a C4.5 decision tree, a linear discriminant analysis, and an extreme learning machine, are used to predict early CKD. The study includes datasets from 19,270 patients, provided by an adult health examination program from 32 chain clinics and three special physical examination centers, between 2015 and 2019. There were 11 independent variables, and the glomerular filtration rate (GFR) was used as the predictive variable. The C4.5 decision tree algorithm outperformed the three comparison models for predicting early CKD based on accuracy, sensitivity, specificity, and area under the curve metrics. It is, therefore, a promising method for early CKD prediction. The experimental results showed that Urine protein and creatinine ratio (UPCR), Proteinuria (PRO), Red blood cells (RBC), Glucose Fasting (GLU), Triglycerides (TG), Total Cholesterol (T-CHO), age, and gender are important risk factors. CKD care is closely related to primary care level and is recognized as a healthcare priority in national strategy. The proposed risk prediction models can support the important influence of personality and health examination representations in predicting early CKD.

## 1. Introduction

Chronic kidney disease (CKD) is a global public health problem and is related to serious morbidity, mortality, and health resource utilization. In 2017, the number of cases worldwide was 69.75 million, and CKD caused 1.2 million deaths. The global prevalence of CKD was 9.1% in 2017. According to the Taiwanese Ministry of Health and Welfare’s annual report, CKD accounts for the largest number of health insurance claims, with 364,000 admitted patients, costing approximately NTD (New Taiwan Dollar) $51.3 billion in 2018. With an aging population and the associated increasing prevalence of hypertension, hyperlipidemia, and hyperglycemia, the number of CKD patients has increased continuously.

Early CKD has no obvious symptoms. A CKD patient’s renal function gradually declines, and uremia develops; at this stage, the patient must receive dialysis or kidney transplantation. Two standards define CKD: (1) the kidney has been injured for over three months, including structural and functional abnormalities, some other way to address what appears to be pathological abnormalities, blood, urine, or imaging abnormalities, and (2) glomerular filtration rate (GFR) < 60 mL/min/1.73 m^2^ for over three months. In general, CKD is divided into five stages based on estimated GFR (eGFR) [1] (Table 1).

Current screening procedures for CKD are inadequate at detecting early CKD [2,3]. In Taiwan, there are at least 2 million CKD patients; however, only 3.5% of them have been diagnosed and informed. Detecting chronic renal failure is difficult until 25% of renal function has already been lost. Early diagnosis can possibly prevent or dampen CKD progression to end-stage renal disease [4].

This study was designed to identify CKD risk factors via Taiwanese adult preventive health examination data for early prediction of decreased kidney function. Since 2012, Taiwan has implemented the “five-year plan for chronic kidney disease prevention and enhancing the quality of care, 2012–2016”. The program’s outcomes included reduced dialysis incidence and increased five-year survival rate of patients after kidney transplantation. However, in 2017, Taiwan reported 275,000 CKD cases and 6743 CKD deaths [5].

In consideration of the heterogeneity of CKD deterioration, it is critical to conduct risk assessment, monitoring, and prognosis from an evidence-based medical viewpoint. A recent survey showed a high prevalence of CKD among the Taiwanese population, with an alarmingly low awareness rate. Moreover, CKD risk factors, such as high blood pressure, low socioeconomic status, and herbal medication, are common in Taiwan. Predictive factors for CKD have been examined extensively in recent years, but remain controversial [6,7,8,9]. Based on a report from the US Preventive Services Task Force (USPSTF) and the American College of Physicians (ACP), CKD screening in asymptomatic individuals is insufficient, and there are no valid tools for CKD screening [9].The American Society of Nephrology strongly recommends regular screening for CKD, regardless of risk factors [10].

It is well known that the bidirectionality plays a critical role in dyslipidemia and proteinuria, and also affects lipoprotein metabolism [11]. The average values of HDL-L (High-Density Lipoprotein Cholesterol) and LDL-C (Low-Density Lipoprotein Cholesterol) are lower in stage 3 to stage 5 CKD patients than in healthy individuals [12]. Chronic renal failure is associated with many factors, including hypertension and proteinuria. For example, it is well known that the magnitude of the blood pressure (hypertension) reduction appeared greater with the progression of CKD. In contrast to early CKD, it has been reported by many studies that hypertension is a comorbidity of CKD, but less studied in early CKD [13,14,15]. Because of CKD’s heterogeneity, the answers to screening and clinical practice are not clear. However, an accurate tool to predict CKD is urgently required. Early CKD awareness is essential for potential patients to participate in and comply with adult preventive health examination programs. Indeed, data mining has been successfully used for building a predictive model for healthcare prediction tasks [16,17,18,19,20]. Thus, in this study, four data mining algorithms, including a classification and regression tree (CART), a C4.5 decision tree, a linear discriminant analysis (LDA), and an extreme learning machine (ELM) are used to predict early CKD. Specially, this study aimed to utilize four data mining methods. In addition, these methods have the potential to explore important risk factors of early CKD and interpretation of the association between each other.

## 2. Materials and Methods

### 2.1. Data Source

All samples were taken from an adult health examination dataset, which are collected from 32 chain clinics and three special physical examination centers. Data from 1 January 2015 to 31 December 2019 were included, giving a total of 19,270 effective records, including 5101 CKD patients and 14,169 non-CKD patients. Personal information, physical examination data, and blood test results from the physical examination database were included, and a total of 11 independent variables were identified. The dependent variable was GFR (Table 2).

### 2.2. Method

This study aimed to utilize four data mining methods involving CART, C4.5, LDA, and ELM to predict early CKD.

CART is a decision tree system which uses a binary recursive procedure to partition the data in homogenous subsets based on the Gini index [21,22]. The partitioning is repeated until the nodes are homogenous enough to be terminal. The first step of CART analysis is building the maximal tree by binary split-procedure, which describes the data. The second step is pruning the overgrown tree and deriving a series of less complex trees from the maximal tree. The last step is to select an optimal tree size using a cross-validation procedure.

C4.5 is also a decision tree algorithm which selects the decision tree’s attributes on each node based on the concept of information entropy. It adopts a greedy approach in which the decision trees are constructed in a top-down, recursive divide and conquer manner [23,24]. At each node of the tree, C4.5 select one attribute by maximum information gain ratio that most effectively splits samples of current node into subsets in one class or the other. The C4.5 algorithm then proceeds recursively until meeting some commonly used stopping criteria, such as the minimum number of samples in a terminal node.

LDA is a well-known generic method used for dimensionality reduction and classification [25,26]. LDA tries to find a low dimensionality space for different categories. In this space, the distances between the samples from different categories are large, but the distances between the samples in the same category are small. In the learning process, LDA can obtain a function to project the samples from different categories onto the low dimensionality space. It applies an eigendecomposition on the scatter matrices to compute the optimal projection.

Based on the projection, LDA can derive a classification model which focuses on the association between multiple independent variables and a categorical dependent variable by forming a composite of the independent variables.

An ELM is a computationally efficient neural network model with a non-iterative learning strategy [27,28]. It randomly selects the input weights and analytically determines the output weights of the neural network. The modeling time of ELM is faster than traditional network learning algorithms, such as the well-known back-propagation neural network. It also reduces many of the difficulties in parameter setting, including stopping criteria, learning rate, and learning epochs, etc. The CART prediction model was built using the *raprt* R package of version 4.1.15 (R core team, Vienna, Austria) [29]. To search the best parameter set to generate a promising CART model, the *OptimClassifier* R package of version 0.1.5 (R core team, Vienna, Austria) was implemented for the parameters of tree depth, number of observations in any terminal node, and tree pruning [30]. To build the C4.5 model, the *RWeka* R package of version 0.4–42 (R core team, Vienna, Austria) was applied [31]. To find the best parameter set for the cost to build an effective C4.5 model, the *caret* R package of version 6.0–84 (R core team, Vienna, Austria) was implemented [32]. LDA was implemented using the *MASS* R package of version 7.3–51.5 (R core team, Vienna, Austria) [33]. The default settings were used to build an LDA model. The ELM model was constructed by implementing the *elmNN* R package of version 1.0 (R core team, Vienna, Austria) [34]. The default activation function in this package is radial basis. To search the best number of hidden neurons that would generate promising ELM models, the *caret* R package of version 6.0–84 (R core team, Vienna, Austria) was used to tune important hyperparameters [32]. Classification accuracy was evaluated using receiver operating characteristic curve analysis to estimate the area under the curve (AUC). Accuracy, sensitivity, and specificity were considered in this study.

## 3. Results

In this study, we applied machine learning approaches to an adult health examination dataset to predict patients with high CKD risk based on the data of each variable for each patient. Our aim was to compare different classification models and identify the most efficient.

Subject demographics are outlined in Table 3. The independent variables in the analysis were gender, age, red blood cell count (RBC), fasting glucose level (GLU), triglycerides (TG), total cholesterol (T-CHO), High-Density Lipoprotein Cholesterol (HDL-C)), Low-Density Lipoprotein Cholesterol (LDL-C), albumin (ALB), proteinuria (PRO), and urine protein to creatinine ratio (UPCR). The t-test is used to compare the averages of age for CKD and Non-CKD. We utilized the chi-square test to evaluate the associations between the dependent variable and all independent variables except age.

We found that age (*p* < 0.001), gender difference (*p* < 0.001), normal or abnormal performances of RBC (*p* < 0.001), GLU (*p* = 0.004), TG (*p* = 0.011), HDL (*p* = 0.029), PRO (*p* < 0.001), and UPRC (*p* < 0.01) were significantly associated with the prevalence of CKD. The *t*-test results showed that the CKD group’s mean age was significantly different from the non-CKD group. The chi-square test analysis suggested that different genders had dissimilar interference on prevalence of CKD, and the paired comparison revealed that the proportion of males in the CKD group was higher (48.3% vs. 39.6%) than that in the non-CKD group.

A higher proportion of subjects with abnormal RBC was found in the CKD group than the non-CKD group (23.2% vs. 19.1%), and a higher proportion of normal GLU was found in the CKD group than the non-CKD group (20.7% vs. 18.8%).

The CKD group contained a higher proportion of subjects with abnormal triglycerides (TG) than the non-CKD group (60.6% vs. 58.5%), as well as a higher proportion of subjects with normal high-density lipoproteins (HDL) (85.6% vs. 84.4%). The CKD group contained a higher proportion of subjects with abnormal proteinuria (PRO) than the non-CKD group (82.1% vs. 35.0%) and a higher proportion of subjects with abnormal UPRC compared to the non-CKD group (67.9% vs. 12.7%). No significant differences were found between normal and abnormal performances of T-CHO (*p* = 0.491), low-density lipoproteins (*p* = 0.782), or ALB (*p* = 0.457).

We randomly selected 15,416 patients (80% of the total patients) as the training samples, while the remaining 3854 patients (20% of the total patients) were employed as the testing sample for measuring out-of-sample predictive ability of the four methods. Moreover, a 10-fold cross validation method was used for training the classification models of the four method.

Table 4 shows the classification results of the CART, ELM, C4.5, and LDA methods. It shows that the AUC values of the CART, ELM, C4.5, and LDA models were 0.779, 0.692, 0.788, and 0.773, respectively. The C4.5 model provided the highest AUC value, followed by the CART, the LDA, and the ELM model, respectively. The accuracy, sensitivity, and specificity values of the C4.5 model are all greater than the three competing models. Figure 1 shows the ROC curves of the four classification methods for the occurrence of early CKD. This figure also depicts that the C4.5 method showed the best predictive ability compared to the three comparison models and is a promising method for early CKD prediction.

## 4. Discussion

The goal of the analysis was to identify the most important risk factors from ten potential factors: gender, age, RBC, GLU, TG, T-CHO, HDLC, LDLC, ALB, PRO, and UPCR. Our results revealed that the C4.5 method can generate the best classification and most promising results to predict CKD. The C4.5 method automatizes the detection of associations between predictors and outcomes and the interactions among predictors and provides metrics of predictor importance. Figure 2 shows the classification tree of CKD predictors using the C4.5 method. Table 5 shows the summarized rules of condition variables.

Subjects were divided into 11 subgroups, from root node to leaf nodes, through different branches. As previously explained, the UPCR variable has a great influence on the interpretation of the eGFR value and was, therefore, identified as the root node of the classified decision tree. The first-level decision tree was obtained from the determining factor: UPCR. The accuracy (ACC) obtained was 88.3% across the 11,189 samples. It means that out of 11,189 non-CKD patients, 9879 patients were correctly classified using UPCR variable. The second-level decision tree was generated by the following factors: PRO, age, RBC, GLU, TG, T-CHO, and gender. Therefore, the decision tree can be divided into abnormal (ABNL; CKD) or normal (NL; non-CKD) situations. The accuracy ranged from 58.3% to 88.3%.

The second-level decision tree was obtained from the following determining factors: UPCR (with ABNL) + PRO (with NL), and the accuracy obtained was 0.749 across 383 samples. The third-level decision tree was obtained from the following determining factors: UPCR (ABNL) + PRO (ABNL) + age (>65.45), and the accuracy obtained was 0.799 across 2285 samples. The fourth-level decision tree was obtained from the following determining factors: UPCR (ABNL) + PRO (ABNL) + age (<65.45) + RBC (ABNL), and the accuracy obtained was 0.757 across 345 samples. The left-hand fifth-level decision tree was obtained from the following determining factors: UPCR (ABNL) + PRO (ABNL) + age (<65.45) + RBC (NL) + age (<51.95) + GLU (ABNL), and the accuracy obtained was 0.685 across 286 samples. Meanwhile, for the following determining factors: UPCR (ABNL) + PRO (ABNL) + age (<65.45) + RBC (NL) + age (<51.95) + GLU (NL) + TG (ABNL), the accuracy obtained was 0.682 across 22 samples.

The right-hand fifth-level decision tree was obtained from the following determining factors: UPCR (ABNL) + PRO (ABNL) + age (<65.45) + RBC (NL) + age (<51.95) + TG (NL) + T-CHO (ABNL) + gender (male), and the accuracy obtained was 0.682 across 22 samples. Meanwhile, for the following determining factors: UPCR (ABNL) + PRO (ABNL) + age (<65.45) + RBC (NL) + age (<51.95) + TG (NL) + T-CHO (ABNL) + gender (female), the accuracy obtained was 0.725 across 91 samples. By using these different decision tree models, clinicians can identify the combinations of factors for a condition of interest.

The findings of this study were consistent with those of previous reports, including the most recent report of the National Health Research Institutes Annual Report on Kidney Disease in the urine protein to creatinine ratio (UPCR) [4] and the red blood cell count (RBC) [5], and Xiao’s report on the prediction of chronic kidney disease in proteinuria (PRO) [35]. The findings of the albumin (ALB) and fasting glucose level (GLU) are consistent with previous studies following Korbut et al. [36] and Kshirsagar et al. [37]. Similarly, Xue et al. [38], Mahmood et al. [39], and Kshirsagar et al. [37] reported that triglycerides (TG), age, and gender are critical for prediction of chronic kidney disease. As suggested from our results, the main issue is how to predict CKD who are asymptomatic and who only undergo a routine adult health examination program. A comprehensive, clinical approach to prevention that considers all of these factors is therefore required to successfully tackle and specifically target the high-risk exposures in the adult population.

Optimal preclinical management of early CKD would therefore benefit from better understanding of the nature. Many of the risk factors that are possibly associated with early CKD awareness, i.e., management of hypertension, are interesting and warrant further investigation.

The empirical results showed that C4.5 slightly outperformed the CART and LDA methods. But, as our work was to explore important risk factors of early CKD and discuss the association between each other, the results of the best method with promising performance is the most suitable for the further discussion. Thus, the classification tree depicting CKD predictors of C4.5 is discussed in this study. Using different kinds of CKD data to compare the effectiveness of C4.5, CART, and LDA for the prediction of early CKD can be considered as one of future research directions.

This study used ELM, C4.5, CART, and LDA to predict early CKD. LDA is a statistical method. The important characteristic of LDA is that LDA projects the data onto a lower dimensional vector space, which is a more discriminant sub-space since the ratio of the between-class distance to the within-class distance is maximized. ELM is a neural network method. Its main characteristic is that the parameters of hidden layers are randomly generated independent of training samples without fine-tuning, thus it has faster learning speed compared with the traditional neural network learning algorithms. The CART and C4.5 are both decision tree methods. The characteristic of a decision tree is to use a set of “if-then” conditions to perform classification of cases. The main feature of CART is that it generates only binary trees based on the Gini impurity index. The C4.5 used gain ratio as the goodness measure to generate decision trees which include multi-branch (i.e., not only binary) splits at each node. However, the ELM model performed poorly in this study. ELM is a neural network algorithm and its modeling mechanism is different from that of the other three methods, C4.5, CART, and LDA. Neural network algorithms are powerful tools for clinical data analysis for prediction of an outcome. Actually, it may be useful but the neural network will not give us any insight information in this study as it cannot be used to select important variables.

## 5. Conclusions

Chronic kidney disease (CKD) is a major global public health problem, but early-stage diagnosis is problematic due to asymptomatic presentation. Currently, there are no widely accepted predictive instruments for early CKD; therefore, physicians must make clinical decisions about which patients to treat. In this study, our aim was to explore important risk factors of early CKD and discuss the associations between them. Importantly, early CKD awareness is essential for potential patients to participate in, and comply with, health examination programs, and is of great clinical and economic significance.

Moreover, to the best knowledge of the authors, there are no studies using data mining classification techniques for building predictive models for early CKD prediction tasks. In this study, we applied nine physical examination variables and two demographic parameters to determine CKD risk factors using four data mining algorithms. The C4.5 algorithm yielded an output of eight features that were important for early CKD prediction. It was found that reducing the number of features increased the accuracy of the results. Another important finding of this paper was that the C4.5 method had the best predictive ability compared to the other three comparison models. C4.5 also revealed that different combinations of dataset attributes resulted in different accuracy rates ranging from 59.6% to 88.3%. We also identified that UPCR, PRO, age, RBC, GLU, TG, T-CHO, and gender had important impacts on the predictivity of the models, while other predictors, such as HDL, LDL, and ALB, were less important.

With the slow progress of CKD, early detection and effective treatment are the only ways to reduce mortality. Timely risk assessment of CKD and appropriate community monitoring are important for preventing further kidney injury in early CKD patients. In conclusion, this work presents evidence of the applicability of an adult health examination dataset and the robustness of the four models for clinical risk assessment of early CKD.

## Figures and Tables

**Figure 1 ijerph-17-04973-f001:**
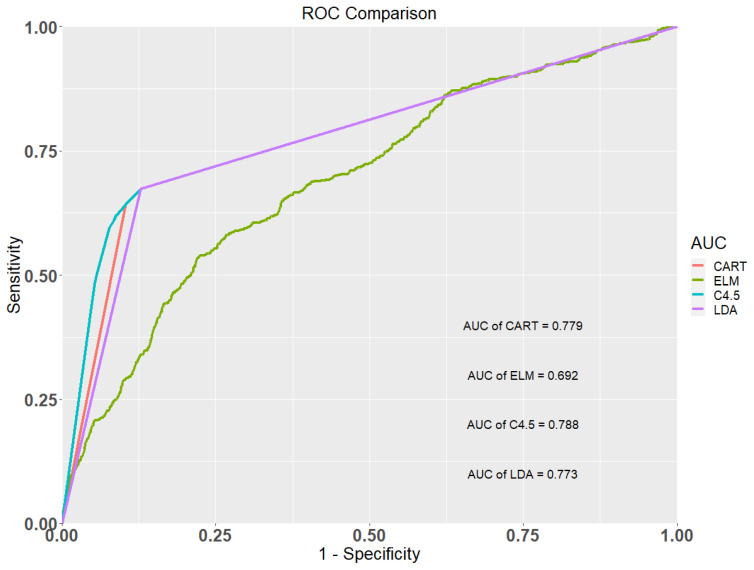
Receiver operating characteristic (ROC) curves of the four methods.

**Figure 2 ijerph-17-04973-f002:**
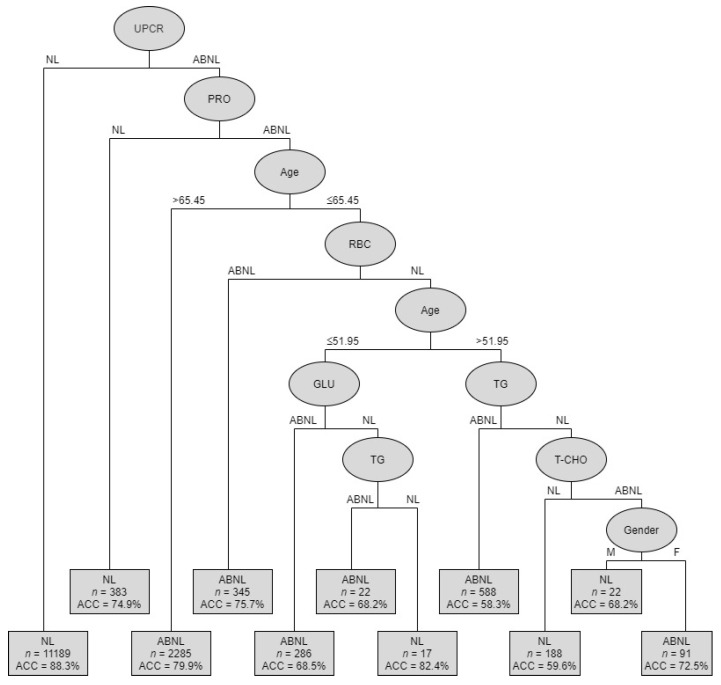
Classification tree depicting CKD predictors of C4.5.

**Table 1 ijerph-17-04973-t001:** The Stages of chronic kidney disease (CKD).

Stage	Description	Estimated GFR
1	Kidney damage with normal or increased GFR	≥90 mL/min/1.73 m^2^
2	Kidney damage with small decrease in GFR	60–89.9 mL/min/1.73 m^2^
3	Kidney damage with moderate decrease in GFR	30–59.9 mL/min/1.73 m^2^
	3a	45–59.9 mL/min/1.73 m^2^
	3b	30–44.9 mL/min/1.73 m^2^
4	Kidney damage with large decrease in GFR	15–29.9 mL/min/1.73 m^2^
5	Kidney failure with need for dialysis (end-stage renal disease)	<15 mL/min/1.73 m^2^

GFR: Glomerular Filtration Rate; 3a: Stage 3a of kidney disease; 3b: Stage 3b of kidney disease.

**Table 2 ijerph-17-04973-t002:** Important variables and coding in this study.

Variable	Name	Definition of Normal Test Data
X1	Gender	Male/Female
X2	Age	Age greater than 40 years
X3	Red blood cells (RBC)	0–5
X4	Glucose Fasting (GLU)	70–100
X5	Triglycerides (TG)	50–150
X6	Total Cholesterol (T-CHO)	50–200
X7	High-Density Lipoprotein Cholesterol (HDL-C)	>40
X8	Low-Density Lipoprotein Cholesterol (LDL-C)	<130
X9	Albumin (ALB)	3.5–5.0
X10	Proteinuria (PRO)	+/−
X11	Urine protein and creatinine ratio (UPCR)	<150
Y	Glomerular filtration rate (GFR)	≥90 mL/min/1.73 m^2^

**Table 3 ijerph-17-04973-t003:** Subject demographics.

Characteristic	Non-CKD	CKD	*p*-Value
*N* (%)	14,169 (73.5%)	5101 (26.5%)	
**Gender**			
Male	5608 (39.6%)	2465 (48.3%)	<0.001 **
Female	8561 (60.4%)	2636 (51.7%)	
**Age**			
Mean (±SD)	63.37 ± 11.56	69.19 ± 10.74	<0.001 *
**RBC**			
Normal	11,460 (80.9%)	3917 (76.8%)	<0.001 **
Abnormal	2709 (19.1%)	1184 (23.2%)	
**GLU**			
Normal	11,502 (81.2%)	1055 (20.7%)	0.004 **
Abnormal	2667 (18.8%)	4046 (79.3%)	
**TG**			
Normal	5878 (41.5%)	2012 (39.4%)	0.011 *
Abnormal	8291 (58.5%)	3089 (60.6%)	
**T-CHO**			
Normal	9198 (64.9%)	3284 (64.4%)	0.491
Abnormal	4971 (35.1%)	1817 (35.6%)	
**HDL-C**			
Normal	11,954 (84.4%)	4369 (85.6%)	0.029 *
Abnormal	2215 (15.6%)	732 (14.4%)	
**LDL-C**			
Normal	11,400 (80.5%)	4095 (80.3%)	0.782
Abnormal	2769 (19.5%)	1006 (19.7%)	
**ALB**			
Normal	14,162 (100.0%)	5097 (99.9%)	0.457
Abnormal	7 (0.0%)	4 (0.1%)	
**PRO**			
Normal	9203 (65.0%)	915 (17.9%)	<0.001 *
Abnormal	4966 (35.0%)	4186 (82.1%)	
**UPCR**			
Normal	12,364 (87.3%)	1639 (32.1%)	<0.001 *
Abnormal	1805 (12.7%)	3462 (67.9%)	

** *p*-value < 0.01; * *p*-value < 0.05.

**Table 4 ijerph-17-04973-t004:** Classification results of the four methods.

Methods	Accuracy	Sensitivity	Specificity	AUC
Classification and Regression Tree (CART)	0.819	0.670	0.871	0.779
Extreme Learning Machine (ELM)	0.715	0.539	0.777	0.692
C4.5	0.820	0.673	0.872	0.788
Linear Discriminant Analysis (LDA)	0.818	0.669	0.868	0.773

**Table 5 ijerph-17-04973-t005:** Summarized rules of condition variables.

Rules No.	Combinations of Condition Variables	Cases of (Ab)normal		Accuracy
1	UPCR (NL)	9879	NL	88.3%
2	UPCR (ABNL) + PRO (NL)	287	NL	74.9%
3	UPCR (ABNL)+PRO (ABNL) + Age (>65.45)	1826	ABNL	79.9%
4	UPCR (ABNL) + PRO (ABNL) + Age (≤65.45) + RBC (ABNL)	261	ABNL	75.7%
5	UPCR (ABNL) + PRO (ABNL) + Age (≤65.45) + RBC (NL) + Age (≤51.95) + GLU (ABNL)	196	ABNL	68.5%
6	UPCR (ABNL) + PRO (ABNL) + Age (≤65.45) + RBC(NL) + Age (>51.95) + TG(ABNL)	343	ABNL	58.3%
7	UPCR (ABNL) + PRO (ABNL) + Age (≤65.45) + RBC (NL) + Age (≤51.95) + GLU (ABNL) + TG (ABNL)	15	ABNL	68.2%
8	UPCR (ABNL) + PRO (ABNL) + Age (≤65.45) + RBC(NL) + Age (≤51.95) + GLU (ABNL) + TG(NL)	14	NL	82.4%
9	UPCR (ABNL) + PRO (ABNL) + Age (≤65.45) + RBC (NL) + Age (≤51.95) + GLU (ABNL) + TG (NL) + T-CHO (NL)	112	NL	59.6%
10	UPCR (ABNL) + PRO (ABNL) + Age (≤65.45) + RBC (NL) + Age (≤51.95) + GLU (ABNL) + TG (NL) + T-CHO (ABNL) + Gender (M)	15	NL	68.2%
11	UPCR (ABNL) + PRO (ABNL) + Age (≤65.45) + RBC (NL) + Age (≤51.95) + GLU (ABNL) + TG (NL) + T-CHO (ABNL) + Gender (F)	66	ABNL	72.5%

UPCR: Urine protein and creatinine ratio, PRO: Proteinuria, RBC: Red blood cells, GLU: Glucose Fasting, TG: Triglycerides, T-CHO: Total Cholesterol.

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
