# Peer review of "Risk Prediction for Early Chronic Kidney Disease: Results from an Adult Health Examination Program of 19,270 Individuals"

_ijerph, 2020, doi:10.3390/ijerph17144973_

Round 1
Reviewer 1 Report
19: to reduce?
26: predicted?
48: "functional abnormalities; such" or some other way to address what appears to be a run-on
49: define GFR? It's present in the abstract, but CKD for eaxmple is redefined on 37
Table 1: standardize m2 or m^2
75: The bidirectional what?
79: clearly, and
88, 90, 92: were, are, were
95: Perhaps this is a difference in preferences, but "predictive" for me is usually used to mean "has the ability to predict" and would better describe the independent variable. "predicted" would be clearer here to me. You call it "dependent" below on 102, and this is probably best since you're using "independent" elsewhere.
118: low-dimension space in which ...
121: eigendecomposition, one word
134: settings were
Table 3: We are not told which test was applied to obtain the p-values until after the table.
176: depicts that shows seems redundant
199: accuracy (ACC) was... It may be useful to give a brief sentence after this first example. What does this statistic mean again? 88.3% across 11,189 means that 88.3% of those were correctly classified when using just this one variable? Or is it something else?
226: The best-performing algorithm? Which one do you mean here?
227: Does this mean, for example, that when you just use UPCR, the accuracy is higher? It might be useful to expand on this a little bit and give an example.
I do not believe that accuracy necessarily falls - I believe that accuracy on the subgroups falls. Correct me if I'm wrong, but using UPCR gives you accuracy 88.3% when tested on 11,869 samples. PRO gives you accurate results on a smaller set that was originally marked ABNL by the UPCR?
In other words, overall accuracy must increase if you use additional independent variables (or the model is broken), but the accuracy on the subgroups decreases. Therefore, it's useful to use more variables, but the provide less marginal predictivity.
195-212: This is difficult to read all in text form and could easily be formed into a table with accuracy, samples, and variable columns to summarize figure 2.
I would expect to see columns: # of variables included, names of variables included, # classified as normal abnormal at this stage, overall accuracy at this level, accuracy in the subgroup.
232: predictivity of the models...
234: are the only way to
Some note might be made that CART, C4.5, and LDA are nearly equivalent in goodness of fit - a difference in AUC of less than 2% might not even be statistically significant despite the size of this set. It could be as simple as noting that while one model is best, it is nearly equivalent to two others and is chosen for further examination here, while these two others might do just as well in practice. This model is promising, but that does not mean the others are not.
ELM clearly falls out of consideration and acts differently in the ROC, and it would be useful for the authors to consider why the model performed so poorly and differently here. There is something essentially different - it's not slicing up the data with planes or trees - and this explains why it differs so substantially.
Methods: I can generally understand this section, but there are some bits we don't need (e.g. CART here is used for classification, so we do not really need to consider CART used for regression) and some that are undefined in the text.
Generally, it would be best to give a higher-level overview of the functions, reduce the jargon, and refer the reader to paper(s) with details and straightforward examples. We are given lots of terms we do not understand - if they're present, they need to be more useful than confusing.
Very clearly, the authors are technically capable and there is nothing in the analysis to give me pause, but the details here are occasionally confusing and will not permit the reader to reproduce anything like this work. Either more details are given on the way in which the models are fit (which is probably not desirable) or the descriptions need to be simpler.
Reviewer 2 Report
Proposed paper is interesting, however some revision are needed before it can be accepted for pubblication:
- The method used is quite complicated and a brief explanation of differences in alghoritm used is needed also in the introduction.
- As mentioned by the authors, BP values are one of the most important determinants of CKD. It is not present in this analysis and this should be discussed in the relative section and not only inserted into the limitations.
- The clinical use of such a complicated alghoritm need to be explained in the discussion.
Reviewer 3 Report
Thanks for the paper.
This is an important topic given high disease burden and use of health resources. Early identification of chronic kidney disease is important.
Some things for the authors to clarify, as follows:
1 The research used a cross-sectional study design. It looks they used the GFR as the outcome variable for chronic kidney disease. If GFR can be used as the measure for the outcome, why bother to undertake the study if the purpose is just to identify the patients at an early stage? The GFR does not seem hard to know with a lab test. What's the value of the risk prediction for early chronic kidney disease? for heath education and promotion?
2 The research used 4 data mining algorithms using the physical examination data. What would be the performance of the 4 algorithms for external data set? We can't rely on the training data only for identification of the best model/algorithms due to issues such as over-fitting.
3 The 'discussion' is not really a discussion. It is actually results. The authors will have to develop this part properly with reference to international development.
4 The conclusions will need to modified accordingly after revision.
5 Statistical methods for Table 2 should be included. 'Accuracy' should also be defined.
6 The univariate analysis findings are in some cases contrary to the common sense understanding, e.g. the CKD group had a higher proportion of 'normal' GLU than the Non-CKD group. Why is that? Regardless of the univariate findings -significant or not, all these variables should be considered in the algorithm development (to control for con-founding, to understand interaction or other associations between variables).
7 Does 'age' follow normal distribution? If not, it should be reported as median and IQR.
Round 2
Reviewer 2 Report
issue raised has been answered and paper can now be pubblished
Author Response
Thank you for this positive comment.
Reviewer 3 Report
Thanks for making efforts to revise the paper, which has been improved a lot.
There are still a few issues for you to address further:
1 The parts highlighted in yellow in 'Discussion' should mainly go to the 'Results'. The 'Discussion' has had some improvement, but it is rare to have a discussion part in a paper without any published references. You have indicated that the paper is mainly about the predictive factors related to CKD. Where are the international papers talking about it whether it being by a data mining method or a traditional statistical method? I would like the 'Discussion' part to be further enhanced.
2 There should be a corresponding description in '2.2 Method' about what you did in lines 175-158 in the 'Results' - something like you used training sample and testing sample.
3 As you've used testing sample in this study, what's the performance when you used the testing sample in terms of accuracy? The reason why I asked this question is because the data mining methods are susceptible to biases and over-fitting.
4 The grammar needs to be improved. In general, the simple past tense should be used to describe the methods and results. A couple of examples as follows: Line 178, 'are' (which should be 'were'); Line 252 'neural network algorithm' (which should be 'algorithms').
Author Response
1 The parts highlighted in yellow in 'Discussion' should mainly go to the 'Results'. The 'Discussion' has had some improvement, but it is rare to have a discussion part in a paper without any published references. You have indicated that the paper is mainly about the predictive factors related to CKD. Where are the international papers talking about it whether it being by a data mining method or a traditional statistical method? I would like the 'Discussion' part to be further enhanced.
Response: The authors appreciate the reviewer’s comments. We already follow reviewer's comment and have paid more explanation to discuss some papers about predictive factors related to CKD. Please refer to the reference list between [35]-[39] in this revision.
2 There should be a corresponding description in '2.2 Method' about what you did in lines 175-158 in the 'Results' - something like you used training sample and testing sample.
Response: Thanks for your comments. The descriptions in Line 158-175 are the descriptive statistics of all 19270 patients. The methods presented in the 2.2 method section were not used to generate the statistics.
3 As you've used testing sample in this study, what's the performance when you used the testing sample in terms of accuracy? The reason why I asked this question is because the data mining methods are susceptible to biases and over-fitting.
Response: Thanks for your comments. The performances of the four used methods for the testing sample has been shown in Table 4: Classification results of the four methods. The metrics used in Table 4 is not only accuracy, but also Sensitivity, Specificity, and AUC
4 The grammar needs to be improved. In general, the simple past tense should be used to describe the methods and results. A couple of examples as follows: Line 178, 'are' (which should be 'were'); Line 252 'neural network algorithm' (which should be 'algorithms').
Response: Thanks for the suggestion. Corrected as suggested by the reviewer.